# Ultrasonographic Evaluation of Skin Toxicity Following Radiotherapy of Breast Cancer: A Systematic Review

**DOI:** 10.3390/ijerph192013439

**Published:** 2022-10-18

**Authors:** Fatimah Alaa Hussein, Hanani Abdul Manan, Aida W. M. Mohd Mustapha, Khairiyah Sidek, Noorazrul Yahya

**Affiliations:** 1Makmal Pemprosesan Imej Kefungsian (Functional Image Processing Laboratory), Department of Radiology, University Kebangsaan Malaysia Medical Centre, Jalan Yaacob Latif, Bandar Tun Razak, Kuala Lumpur 56000, Malaysia; 2Department of Radiology and Intervensi, Hospital Pakar Kanak-Kanak (Children Specialist Hospital), Universiti Kebangsaan Malaysia, Jalan Yaacob Latif, Bandar Tun Razak, Kuala Lumpur 56000, Malaysia; 3Department of Radiology, Hospital Canselor Tuanku Muhriz, Universiti Kebangsaan Malaysia Medical Centre, Jalan Yaacob Latif, Bandar Tun Razak, Kuala Lumpur 56000, Malaysia; 4Department of Radiotherapy, University Kebangsaan Malaysia Medical Centre, Jalan Yaacob Latif, Bandar Tun Razak, Kuala Lumpur 56000, Malaysia; 5Diagnostic Imaging & Radiotherapy Program, Faculty of Health Sciences, School of Diagnostic & Applied Health Sciences, Universiti Kebangsaan Malaysia, Kuala Lumpur 50300, Malaysia

**Keywords:** breast cancer, radiotherapy, ultrasound, skin toxicity

## Abstract

The present review aimed to systematically review skin toxicity changes following breast cancer radiotherapy (RT) using ultrasound (US). PubMed and Scopus databases were searched according to PRISMA guidelines. The characteristics of the selected studies, measured parameters, US skin findings, and their association with clinical assessments were extracted. Seventeen studies were included with a median sample size of 29 (range 11–166). There were significant US skin changes in the irradiated skin compared to the nonirradiated skin or baseline measurements. The most observed change is skin thickening secondary to radiation-induced oedema, except one study found skin thinning after pure postmastectomy RT. However, eight studies reported skin thickening predated RT attributed to axillary surgery. Four studies used US radiofrequency (RF) signals and found a decrease in the hypodermis’s Pearson correlation coefficient (PCC). Three studies reported decreased dermal echogenicity and poor visibility of the dermis–subcutaneous fat boundary (statistically analysed by one report). The present review revealed significant ultrasonographic skin toxicity changes in the irradiated skin most commonly skin thickening. However, further studies with large cohorts, appropriate US protocol, and baseline evaluation are needed. Measuring other US skin parameters and statistically evaluating the degree of the association with clinical assessments are also encouraged.

## 1. Introduction

Breast cancer is the most common malignancy in women [1]. The incidence rate is 10.4% of all cancers globally [2]. Annually, 2.1 million women are diagnosed with breast carcinoma [3]. Adjuvant radiotherapy (RT) is frequently indicated after breast-conserving surgery [4] and in selected cases after mastectomy [5]. In early-stage breast cancer, radiotherapy significantly reduces tumour recurrence and improves overall survival [6]. Despite advancements in RT techniques, skin toxicity or radiation dermatitis is a common and distressing side-effect. Acute or early skin toxicity (up to 3 months) affects nearly all breast cancer patients with some degree of erythema, oedema, or dry desquamation. A higher RT dose may lead to moist desquamation and ulceration [7]. This causes discomfort, restricts daily activities, and may interrupt treatment sessions [8]. Chronic or late skin toxicity symptoms such as telangiectasia, hypo- or hyperpigmentation, and fibrosis are less common in severity. However, they affect the cosmetic appearance of the breast and impact women’s self-image [9]. The pathophysiology of radiation-induced skin toxicity includes factors such as inducing inflammation and the release of inflammatory mediators [10]. Others involve endothelial cell loss of the skin microvasculature and cell loss of the basal layer of the skin that is responsible for cell division [11].

Conventional fractionation whole-breast or -chest irradiation has been the standard RT protocol in breast cancer (55–60 Gy in 25 fractionated doses over 5–6 weeks). This is associated with dose inhomogeneities and excessive breast tissue irradiation leading to more toxicity [12]. A boost treatment of 10–16 Gy, at 2 Gy per fraction, is usually given to the tumour bed using electrons or mini-tangent photons, depending on the depth of the lumpectomy cavity. With modern RT techniques (three-dimensional conformal radiation therapy (3D-CRT) and intensity-modulated RT (IMRT)) and fractionation schedules (hypofractionation and accelerated partial breast irradiation), radiation-related complications can be reduced, ultimately resulting in shorter treatment time and improved patient quality of life [13,14]. Treatment and patient-related factors that increase the risk of acute or chronic skin toxicity from RT have been studied in the literature. Older age, smoking [15], large breast size or volume, high BMI, previous systemic therapy, conventional fractionation RT [16,17], boost treatment [18], maximum RT dose [19], and black race [20] have been implicated. On the other hand, hypofractionated RT is a treatment-related factor associated with less acute and chronic skin toxicity [21,22].

Skin toxicity is routinely assessed by clinical assessments and grading systems (visual observation and palpation) [23]. The Radiation Therapy Oncology Group/European Organisation for Research and Treatment of Cancer (RTOG/EORTC) toxicity criteria and National Cancer Institute Common Toxicity Criteria for Adverse Events (NCI CTCAE) systems are the most widely used [24,25]. They score the toxic effects from grade 0, which means no change, to grade 5, which means death due to these effects. However, these assessments are subjective, with high inter- and intraobserver variability [16,26].

Ultrasound is objective, safe, widely available, and cost-effective, making it a suitable technique in clinical practice compared to other imaging techniques such as MRI and CT. This technique has been used in various dermatologic conditions with excellent reproducibility [27] and treatment response monitoring [28]. It can also be helpful in assessing skin toxicity from head and neck cancer radiotherapy [29,30]. In addition, it can be used to support the development of new treatments for the growing numbers of breast cancer survivors due to improvements in screening, diagnosis, and treatment. The review question was the following: “What are the ultrasonographic changes associated with skin toxicity following RT of breast cancer?” Thus, this systematic review aimed to identify the best evidence on the ultrasonographic skin toxicity changes following breast cancer radiotherapy.

## 2. Materials and Methods

### 2.1. Systematic Review Protocol and Selection Criteria

The systematic review (PROSPERO registration ID: CRD42022328748) was conducted following Preferred Reporting Items for Systematic Reviews and Meta-Analyses (PRISMA) guidelines. Original articles were evaluated on the basis of PICOS criteria (Table 1). The PICOS criteria were followed to develop literature search strategies by systematically determining the inclusion and exclusion criteria based on population, intervention, comparison, outcome, and study design. Studies that fulfilled all five PICOS criteria were included in this systematic review.

### 2.2. Search Strategy and Selection Process

PubMed (National Centre for Biotechnology Information) and Scopus electronic databases were searched to identify relevant articles published between the earliest record and 1 March 2022 with weekly automatic email updates. Search terms used for both databases can be accessed via Appendix A. Research articles were reviewed via title, abstract, and then, finally, via full text by F.A.H and independently reviewed and cross-checked by H.A.M. and N.Y. Reference lists of the included studies were also screened through the Google Scholar database to capture any additional relevant records. Spreadsheet software was used to organise and assess the titles of included studies and identify duplicates, whereas the abstracts were viewed through word-processing software. The selection results were discussed in team meetings until consensus was achieved. The study search and selection were completed on 22 March 2022.

### 2.3. Quality Assessment

We used a quality assessment tool from the National Heart, Lung, and Blood Institute, Quality Assessment Tool for Observational Cohort and Cross-Sectional Studies, to assess the quality of the included studies (Appendix A).

### 2.4. Data Review and Extraction

After finalising the selection process, data extraction was performed by F.A.H and independently reviewed by H.A.M and N.Y. Information was extracted, and the following data were included: author(s), publication year, study country, study design, patient demographic information, US machine, RT treatment protocol, type of breast surgery, type of clinical assessment, and time of evaluations. Then, further data were systematically extracted on the basis of measured skin parameters and locations, ultrasound skin findings, and main findings. Lastly, we summarised the ultrasonographic skin changes according to the time of assessments into early and late skin toxicity changes and their association with clinical assessments.

## 3. Results

### 3.1. Study Selection and Quality Assessment

The database searches yielded 259 records from PubMed and 127 from Scopus. After removing duplicates, 321 articles were reviewed for inclusion via title, abstract, and full text following PICOS criteria. Sixteen studies met the inclusion criteria. In addition, reference lists of the included studies (*n* = 386) were screened, and only one study was included. The literature search process is detailed in the PRISMA flow diagram (Figure 1). The included studies were of moderate quality, except for one of good quality.

### 3.2. Study Characteristics

Table 2 describes the characteristics of the included studies. The selected studies were published between 1998 and 2021 and were geographically diverse, representing eight countries. A total of 6/17 studies were conducted in the same centre in the USA (Emory University School of Medicine), almost by the same group of authors [9,31,32,33,34,35]. One of these studies [9] was a more comprehensive follow-up study with 77% of the patients from the previous one [32]. Lastly, three studies were conducted by the same author in Newcastle Mater Hospital, Waratah, NSW, Australia [36,37,38].

Regarding study design, all of the included studies were prospective. The reports included 783 breast cancer female patients who received radiotherapy. The number, however, may be overestimated due to the possible overlap of patients reported by the same authors or group of authors in different studies, especially researchers from Emory University School of Medicine. The sample size varied widely between 11 and 166 (median, 29). The age range of the reported studies was between 26 to79 years. The BMI (mean and median range 23.5–29.9) of the patients was reported in five studies [9,32,33,39,41], and smoking history was reported in three studies [9,32,41] (*n* = 12/66, 14/70, and 2/34 patients, respectively). In addition, breast volume (range 177–6988.3 cm^3^ or mL) was reported in five studies [9,32,33,41,45].

The majority of the studies included conventional fractionation RT (CF) as a treatment modality as it was the standard protocol before introducing newer techniques such as hypofractionated (HF) RT or accelerated partial breast irradiation (APBI). One study assessed toxicity from HF [40], and two studies compared CF and HF treatments regarding skin toxicity [33,39]. Most of the patients received electron boost technique to the tumour bed. Breast-conserving surgery (BCS) is the most common surgical approach to breast cancer in the included studies. Two studies evaluated patients following BCS and mastectomy [39,44], and one study evaluated patients purely postmastectomy [46].

Most studies used various clinical assessments or scoring to correlate or compare with objective US findings. The most frequently reported were Radiation Therapy Oncology Group (RTOG) and Common Terminology Criteria for Adverse Events (CTCAE) grading scales. The timing of clinical and US assessments varied widely between studies. The studies evaluated acute (early) toxicity, chronic (late) toxicity, or both. Overall, the evaluation time ranged from during RT sessions to 135 months after RT, except for one patient [38] who was evaluated 22 years post RT. A total of 10/17 studies included baseline US evaluation for comparison with subsequent observations and to determine any skin thickening predated RT from operation-related oedema.

### 3.3. Ultrasound Protocol

The studies utilised various brands of ultrasound machines and probe settings, except for three studies, which did not give details about the US machine used [9,32,33]. The probe frequency used for the measurements ranged from 4 to 20 MHz. A total of 8/17 studies performed skin ultrasound with a probe of 20 MHz. Some studies reported the lateral resolution (range 150–200 µm), axial resolution (range 60–70 µm), and maximum depth (range 6–80 mm) of the US probe. The level of the experience of the sonographers was poorly reported in the studies except for four studies: a “radiologist” [40], a “specialist for breast ultrasound” [39], “two specially trained staff” [38], and “two radiation oncologists and one ultrasound expert” [34]. Four studies reported that the sonographer was blinded to the clinical toxicity grading [31,35,40,46]. One of them was also blinded to the patient’s treatment characteristics [40].

### 3.4. US Skin Parameters and Locations of the Measurements

Skin thickness (epidermis plus dermis) was the most frequently measured, which is the distance from the anterior echogenic border of the epidermis to the posterior echogenic border of the dermis using B-mode. Four reports measured the epidermal thickness (the most superficial layer of the skin), one of them as a mean value [36] and the others as skin thickness ratio (STRA) by dividing the mean epidermal thickness of the treated breast by the mean epidermal thickness of the contralateral breast [9,32,33]. Another three studies measured the dermal thickness, which is the middle layer of the skin [41,44,45]. The total cutaneous thickness (epidermal and dermal thicknesses) was another parameter used by [38] to evaluate the presence of cutaneous oedema. Details of the heterogeneity are tabulated in Table 3.

Three studies evaluated the echogenicity of the dermis and the visibility of the echogenic line between the dermis and subcutaneous fat [38,44,45]. The entry echo of the skin and the signal intensity of the dermis were only assessed by [44].

Four studies (almost by the same group of authors) utilised ultrasound radiofrequency (RF) signals to obtain skin thickness as the distance between backscattered signals from the epidermis and those of the hypodermis [31,34,35,42]. In addition, the same group assessed the hypodermal integrity by measuring the Pearson correlation coefficient (PCC) from the US RF data. PCC was obtained by measuring the correlation between two variables representing the adjacent scan lines within a region of interest (ROI) situated along the hypodermal surface. One of these studies extracted the dermal toxicity (the difference between the skin thickness of the treated breast and that of the untreated breast) and the hypodermal toxicity (the difference between 1 minus the PCC of the hypodermal surface on the treated breast and the untreated breast) [34]. All thickness measurements were in millimetres, except for one study calculating the dermal thickness in micrometres [44].

The locations of the US measurements within the breast varied between studies. Four reports measured the four quadrants of the breast [9,32,33,45]. Another four used the 12:00, 3:00, 6:00, and 9:00 positions around the nipple [31,34,35,39]. Three studies by the same author measured the parameters 4 cm medial and lateral to the nipple [36,37,38]. Two reports obtained measurements from the irradiated breast and boost region [40,41]. Other locations were measured from nine points within the medial, central, and lateral areas of the breast [46], the border between the upper quadrants 2–3 cm above the mammilla [44], and the upper medial quadrant [43]. One study measured the irradiated breast without specifying the location [42]. Three studies reported measurements of the boost region to find any difference in skin findings with an additional radiation dose relative to the irradiated non-boosted breast [32,34,40,41]. All included studies used the same locations of measurements on the nonirradiated breast for comparison.

### 3.5. Ultrasonographic Skin Toxicity Changes

All studies reported skin toxicity changes of the irradiated breast documented by ultrasonography relative to the nonirradiated breast regardless of the fractionation schedule of RT and timing of skin reactions (early or late) (Table 3 and Table 4, Figure 2). Most studies found significant differences (*p* < 0.05) in irradiated skin parameters compared to nonirradiated skin. Furthermore, all studies reported skin thickening, except a study by [46] which observed skin thinning with more than 1 year follow-up after postmastectomy RT attributed to fibrosis as part of chronic skin reactions. Three studies noticed the most significant difference in skin thickening at 4–6 months post RT [9,36,43]. However, differences during RT and early skin reactions did not reach statistical significance in two studies [36,37]. A study by [44] noticed no significant difference regarding dermis thickness between the early and late skin reactions when compared to each other.

The following US findings were less commonly evaluated by the included studies: 

Warszawski et al. (1998) [44]: Entry echoes—no significant differences between nonirradiated and irradiated skin for early or late reactions.Signal intensity—significant reduction of the signal intensity of the upper and lower corium in the early and late reactions but was more distinct in the early reactions.Reduction of echogenicity—no significant difference between early and late reactions for the upper corium, but, for the lower corium, differences were significant (more distinct in the early reactions).Border structure—no significant difference in the border structure between the dermis and subcutaneous tissue of the irradiated skin compared to the nonirradiated skin.Schack et al. (2016) [45] and Wratten et al. (2000) [38]: The dermis–subcutaneous boundary was less well defined in the treated breast than in the untreated breast.Decreased dermal density or echogenicity (not quantified).

Hypodermal damage is another RT-related toxicity assessed by four studies by measuring the PCC of the hypodermis from RF data [31,34,35,42]. These reports found a significant decrease in PCC in the early and late reactions, except [34], which found significant differences in the early but not the late reactions. They stated that healthy skin has higher PCC, and that fibrosis following RT will reduce the hypodermal integrity and decrease the PCC.

Several studies reported skin thickening prior RT attributable to the axillary lymph node dissection (ALND) as a part of the surgical treatment of breast cancer. The authors of [9,32] conducted two consecutive studies to evaluate the impact of ALND on breast skin thickening during and up to 1 year post RT. They found a persistent increase in skin thickening from baseline until 1 year follow-up after RT due to axillary surgery. Four studies reported that the medial aspect of the breast was thicker than the lateral aspect [36,37,38,46]. The three studies by Wratten et al. also found the medial aspect of the nonirradiated skin was thicker than the lateral aspect.

Only two studies compared the US skin toxicity changes between CF and HF. Despite reporting significant differences in CTCAE score at the end of RT, the authors of [39] found no significant difference in skin thickening at the end and 6 weeks post RT. They stated that the results might be attributable to the fact that most patients developed only mild radiodermatitis, and differences between HF and CF were too small to be detected by the US, whereas Wang et al. (2020) [33] also found no significant differences in STRA during, 12 weeks post, and 1 year post RT. Both studies supported the hypofractionated approach as having better patient-reported and cosmetic outcomes. Additionally, comparing the boosted and non-boosted regions of the breast, the authors of [40] reported no significant difference, and that additional RT dose to the tumour bed would not lead to more fibrosis and increased skin thickness. The authors of [41] found the same measurements, Torres et al. (2016) [32] did not report the measurements or the differences. Lastly, Yoshida et al. (2012) [34] observed a lack of consistency at the tumour bed because of poor visualisation at this site and recommended eliminating the tumour bed location.

### 3.6. Variables Associated with Skin Toxicity

Some reports used statistical analysis to analyse the predictors or variables associated with skin toxicity. The older age group was a significant predictor of increased skin changes at the end of RT relative to baseline [32]. On the contrary, age < 65 years was significantly associated with more severe skin toxicity on bivariate analysis [41]. Breast volume was a common predictor of a greater increase in skin thickening reported in three studies [32,33,41]. On the other hand, none of the studies that collected the BMI of the patients reported that obesity was a predictor of skin toxicity. Previous chemotherapy and concurrent endocrine treatment did not predict more skin changes at the end or 6 weeks post-RT [32]. At the same time, there was no association between breast retraction/cosmetic outcome and previous systemic therapies [33].

Current smoking is another variable that was a predictor for higher baseline STRA [32] but was not a predictor of STRA at 1 year [9]. Moreover, the Caucasian race was found to be a predictor at 1 year [9] but was not at week 6 post-RT [32]. In addition, no association was found between African American race and breast asymmetry post RT [33]. The RT boost technique did not predict more severe skin changes, whether electron boost at 1 year [9] or photon boost at 6 weeks post RT [32]. Supraclavicular nodal irradiation [33] and the time interval between surgery and RT [9] were also predictors for more severe skin changes 1 year post RT. Interestingly, Wratten et al. (2007) [36] found that the type of node dissection, nodal irradiation, and postoperative wound infection were the most important factors that influenced cutaneous oedema over time using GEE (generalized estimating equations) analysis. They noted that patients who did not have a level 2 node dissection, infection, or regional nodal irradiation demonstrated no increase in epidermal thickness throughout the entire study period.

### 3.7. Association of US Skin Changes with Clinical Assessments

A total of 14/17 studies used various clinical assessments to compare the US skin measurements with clinical evaluations or scales (Table 4). Generally, depending on the time of clinical and US assessments, all of these studies reported a variable degree of association with clinical assessments except for two studies [39,43]. The US skin measurements were higher for patients with more severe visible or palpable skin reactions (higher grades) than patients with mild or no reactions. The reports that compared the early US skin changes revealed that US skin measurements were more significant with increasing clinical grading [32,41] or obvious skin changes [37] than those with less skin changes. However, the authors of [39], when comparing HF and CF groups at the end of RT, reported significant differences in the CTCAE scores but no significant difference in the US changes or symptoms measured by the Skindex-16 questionnaire.

On the other hand, reports comparing late US skin changes found differences in the association pattern between parameters. Of these studies, the authors of [34,45,46] stated that US measurements were most marked with increasing toxicity grading. Wong et al. (2011) [46] used retrospective acute toxicity grading. A study by [40] found a significant direct correlation with higher grades, while Yoshida et al. (2011) [35] found that PCC correlated with RTOG, but skin thickness did not. In addition, they reported no correlation between US measurements and erythema/melanin indices measured by spectrophotometry. Another study from the same group reported that skin thickness correlated with RTOG late subcutaneous toxicity, and PCC correlated with late skin toxicity [31].

Of the studies that compared both early and late skin changes, Wang et al. (2020) [33] documented a significant association between STRA and breast asymmetry or retraction measured by percentage breast retraction assessment (pBRA), while Wratten et al. (2000) [38] found that the most significant thickness was in patients with more prominent visible breast oedema. On the contrary, Keskikuru et al. (2004) [43] did not report any significant correlation in acute and chronic changes. Instead, they found a significant correlation between skin thickness and procollagens (PINP and PIIINP) measured from suction blister fluid of the irradiated skin. They assumed radiation-induced oedema manifested as skin thickening is associated with increased collagen synthesis. A study by [44] noted discrepancies between US changes and RTOG grading in the late reactions, but ultrasonic evaluation could record the structural changes in the early skin reactions much earlier than visible reactions by the naked eye.

### 3.8. US Reliability/Reproducibility

The reliability (intra- and interobserver reliability) of the US was assessed by only one study for evaluating the dermal and hypodermal toxicities from RT using the intraclass correlation coefficient (ICC) [34]. They found that the dermal toxicity parameter was highly reliable (high ICC) while the hypodermal toxicity parameter was moderately reliable. Moreover, only one study evaluated the reproducibility of the US measurements by three operators [32]. They observed no significant inter- or intra-operator differences between measurements compared to the healthy breast at all time points. This finding was a basis for the follow-up study by [9].

## 4. Discussion

Our interest in the present study is to systematically review the skin toxicity changes following breast cancer radiotherapy using ultrasonography. The reports revealed ultrasonographic skin toxicity changes in the irradiated breast compared to the nonirradiated breast. This study is the first systematic review summarising the available evidence for evaluating skin toxicity following breast radiotherapy using ultrasonography. In general, the results of this review demonstrated significant skin toxicity changes during and after radiation, even several years after treatment, relative to the untreated or pre-RT breast measurements. However, nonsignificant changes during and shortly after RT were reported in two studies by the same author.

Despite heterogeneity in the parameters tested and locations imaged, skin thickening was the consistent finding across the studies except one that reported skin thinning after 1 year of pure post-mastectomy RT. The oblique incident angle and flat chest wall may be responsible for increasing the RT dose delivered to the skin leading to thinning, which may explain the increased breast reconstruction complications in postmastectomy patients receiving RT [46]. Further studies with longer follow-ups are needed to document this unusual finding after postmastectomy RT. Radiation-induced skin thickening is somewhat attributed to radiation damage to skin microvasculature resulting in ischaemia and oedema [43]. Nevertheless, a considerable number of studies reported skin thickening before RT as a result of axillary surgery. The axillary surgery disrupts the lymphatic circulation, resulting in lymphatic fluid accumulation, oedema, and breast skin thickening before RT. At the same time, radiation-induced oedema cannot decompress in a patient with disrupted lymphatics secondary to surgery; this increases skin thickening with short [32] and long-term follow-up after RT [9]. Similarly, when the lymphatic drainage of the breast is compromised by surgery, irradiation, or even postoperative wound infection, RT will aggravate the oedematous skin changes and thickening and exert a synergistic effect [9,32,36]. With a longer duration between surgery and RT, there is more time to develop fibrosis from surgery resulting in more severe skin thickening [9]. Axillary irradiation may be a better alternative to ALND as a treatment approach to positive axillary lymph nodes to reduce skin thickening [32]. For future studies, baseline US skin assessment and optimal subgrouping between patients with or without axillary surgery are strongly recommended to enable better quantifying the magnitude of change attributed to RT and allow appropriate comparison.

Another interesting observation in this review is that the medial aspect of the irradiated and even the nonirradiated breast was thicker than the lateral aspect. This can be attributed to the lymph drainage from the medial parts is predominantly through the axilla, while some drainage of the untreated breast also occurs through the axilla of the treated breast. ALND will result in more oedema and increased thickening on the medial side [36]. However, it was reported by just four studies, three of which were by the same author. Future work should document this finding by measuring the same points to assess changes over time.

Limited studies have evaluated other US skin toxicity parameters such as echogenicity and signal intensity of the dermis, entry echo, and visibility of the dermis-subcutaneous fat interface. These studies reported decreased dermal echogenicity and were most distinct in the early skin reactions [44]. However, the echogenicity depends on several factors such as the thickness of the tissue between the transducer and measured point, the echogenicity of the tissues lying at, superficial, and deep to that point, and importantly on the gain setting of the US machine used [38]. Therefore, all these factors should be considered for accurate measurements of dermal echogenicity. Additionally, they identified poor visibility of the dermis–subcutaneous fat boundary, although nonsignificant differences between irradiated and nonirradiated skin were observed by [44]. The increasing gain setting will overcome this boundary’s poor visibility to provide accurate skin thickness measurements [38]. We are undergoing a prospective cohort study to evaluate these parameters; hopefully, we can contribute further evidence for evaluating this common and distressing side-effect of RT.

Moreover, minimal studies assessed the skin toxicity by the US from RT techniques other than conventional fractionation-whole breast irradiation(CF-WBI), which has been blamed for a higher level of toxicity concerning other newer techniques. This shows that most studies were published more than 5 years ago. This issue may be explained that the CF-WBI was the standard radiation schedule at that time that is associated with high radiation doses and damage to the normal tissues in the treated field. To date, only three studies evaluated skin toxicity following HF ultrasonographically, two of which compared CF and HF [33,39]. Both reports supported that HF has better early and late patient-reported outcomes. In particular, as the application of hypofractionation increases, more new RT protocols are being tested in adjuvant WBI prospective trials, aiming for fewer side-effects and shorter treatment time to decrease the burden on breast cancer patients. Therefore, it could be essential to have a quantitative, feasible, and reproducible tool for assessing skin reactions not susceptible to intra- and interobserver variation in adjunct to the physical examination. Thus, more studies are needed to evaluate skin toxicity from emerging techniques such as hypofractionation, partial breast irradiation, and Mammosite.

There were limited observations regarding the skin finding differences between boosted and non-boosted regions of the breast. However, this review shows that a boost dose to the lumpectomy cavity does not contribute to more skin toxicity changes observed ultrasonographically. This finding supports the evidence that boost dose has no to limited impact on long-term cosmetic outcomes [15,18]. Nevertheless, we recommend considering these observations with future work to identify whether adding further RT dose leads to more toxicity changes. At the same time, we encourage assessment of the effect of the type of boost treatment, electron or photon, in separate studies on skin toxicity changes by ultrasound.

Several variables have been studied in the literature, including patient, tumour, and treatment-related factors that predict or associate with increased skin toxicity. These factors appeared to have a greater effect on aggravating or increasing skin changes. Consequently, poor cosmetic outcomes might result with more severe skin changes. A study by [36] found that even RT did not induce skin thickening measured with the US without axillary surgery, irradiation, or postoperative wound infection. This is a very significant observation that needs further investigation. In our review, limited studies assessed or controlled these variables. Across these studies, breast volume was the constant patient-related factor linked to enhanced skin toxicity from RT. Patients with large breast volume have a higher percentage of adipose tissue within the breast that will be more susceptible to RT toxic effects leading to more skin toxicity changes [33]. Other variables studied in our review such as age, smoking, BMI, race, systemic therapy, the time interval between surgery and RT, and nodal irradiation fluctuate in their association with skin toxicity. Further studies are necessary to confirm and control the effects of these predictors.

Despite the subjectivity of the clinical assessments and scoring scales, they are still the commonest toxicity evaluation during and following RT. Comparison with clinical assessments should be considered for any objective/quantitative technique [34]. Most studies reported that the US skin changes were consistent or associated with clinical assessments regardless of the time of evaluations. A significant correlation with procollagens (PINP and PIIINP) measured from suction blister fluid of the irradiated skin is further evidence of the ability of the US to measure skin toxicity changes accurately [43]. Yet, across the studies, the strength of the association (the use of *p*-value or Rho factor) has not been studied well to reach a definitive conclusion. US can also detect skin changes earlier than or even not detected by the naked eye [38,44], allowing for earlier detection or prediction of skin toxicity. Moreover, it may also become useful for assessing new interventions that reduce skin toxicity.

It is essential to consider some issues in the study methodology. First, there was significant variation in sample size across the studies, with six having small sample sizes (<20), which may have affected the overall significance of the skin changes. Second, there was some overlap of patients between studies, especially those assessed by the same authors or centres. This may have reduced the total number of patients evaluated ultrasonographically to reach an accurate conclusion about US skin changes. We are hopeful to see more publications with large cohorts and different centres in the future. Third, it is noteworthy that different ultrasound machines used and inadequate probe frequency (<18 MHz) for skin evaluation utilised by many studies may have contributed to some variability. In addition, the timeframe of assessments varied widely from actual RT sessions to 135 months after RT except for one patient evaluated 22 years post RT. This may have affected time-dependent changes, as noted by [44], which stated that US skin changes depend on the time interval between completion of RT and US assessment. The reliability and reproducibility of the ultrasound measurements were only investigated in two studies. Furthermore, none of the US assessors were blinded to the patient’s radiation exposure, although some studies reported blinding to the clinical grading or the patient treatment characteristics. These factors may have given rise to some limitations or biases.

The ultrasound examination is generally objective, feasible, safe, inexpensive, and widely available. Ultrasound may provide useful development in the noninvasive assessment of RT-related skin toxicity in clinical practice and research settings. However, Wratten et al. (2007) [36] described the use of HFUS mainly in a research setting when assessing interventions that aim to reduce breast oedema, while Schack et al. (2016) [45] stated that HFUS evaluation of the skin is not considered part of large-scale follow-up routines in assessing radiation-induced morbidity. Emerging ultrasound-based techniques, such as elastography, may provide more accurate and objective features to ultrasound B-mode by measuring skin elasticity. It has been measured in different skin diseases [47,48] and gives valuable addition to the US evaluation.

## 5. Conclusions

Skin toxicity post radiotherapy treatment includes skin thickening, less echogenic dermis, a poorly visible dermis–subcutaneous fat boundary, and decreased PCC of the hypodermis compared to the nonirradiated skin. However, further studies with large cohorts and appropriate methodology are encouraged. In addition, future work on measuring other US toxicity parameters is warranted. Furthermore, US evaluation of skin toxicity from newer RT protocols, taking baseline measures, and further grouping patients with risk factors for skin toxicity will provide a more comprehensive assessment of the effect of RT on skin toxicity. Lastly, measuring skin elasticity by ultrasound elastography will further support the ability of the US to measure skin toxicity changes.

## Figures and Tables

**Figure 1 ijerph-19-13439-f001:**
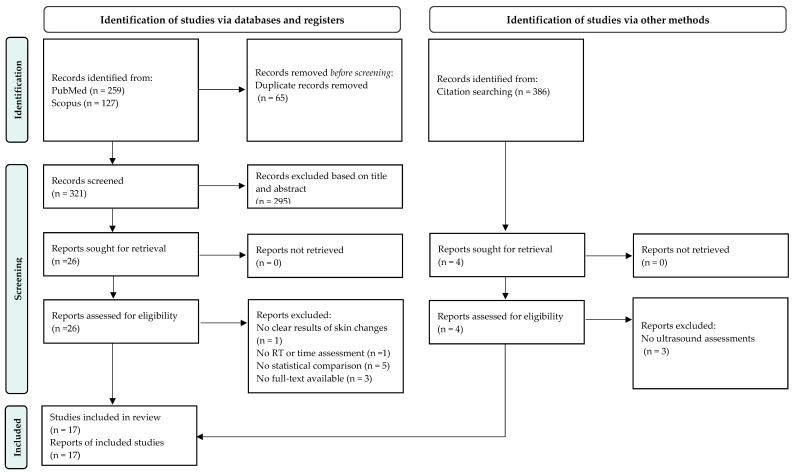
Search strategy based on PRISMA flow diagram for new systematic reviews which included searches of databases, registers, and other sources.

**Figure 2 ijerph-19-13439-f002:**
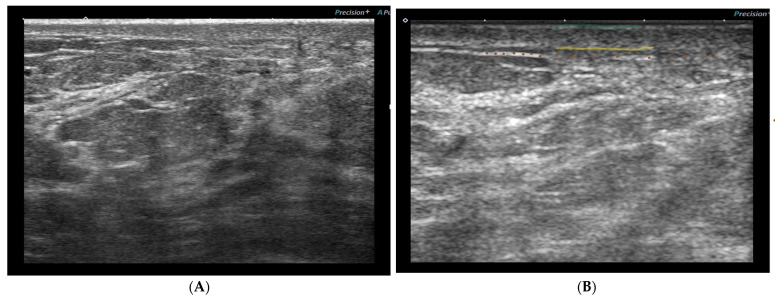
(**A**) An ultrasound breast image of a patient (from our prospective cohort study) before radiation shows some skin thickening. (**B**) The same patient after 15 sessions of RT showing increased skin thickening (from the epidermis (green line) to the lower border of the dermis (yellow line)), decreased echogenicity of the dermis (the layer between green and yellow line), and disturbed deeper echogenic line (brown dots).

**Table 1 ijerph-19-13439-t001:** PICOS criteria for inclusion in the systematic review.

Criteria	
P—patient	Women with breast cancer ≥18 years old
I—intervention	Radiotherapy
C—comparison	Ultrasound assessments of the irradiated skin compared with the contralateral nonirradiated or pre-RT skin measurements
O—outcome	Skin toxicity
S—the type of study	Exclusion of studies with no statistical comparisons (case study or case series) or consisting of fewer than 10 patients, as well as reviews, editorials, and non-English or nonhuman studies

Abbreviations: Pre-RT = pre-radiotherapy.

**Table 2 ijerph-19-13439-t002:** Description of studies utilising ultrasonography to assess skin toxicity following breast cancer radiotherapy.

Author(s), Year, and Country	Study Design	Demographic Information	US Machine	RT Delivery Protocol	Type of Breast Surgery	Clinical Assessment	Time of Clinical and US Assessments
(Borm et al. 2021) [39] Germany	Prospective	Total patients: 166, after hierarchal clustering: 80 Mean age: 58.6 Mean BMI: 24.9	Philips EPIQ 7 w Probe: 18–4b MHz linear array	CF: 50.4 Gy, (*n* = 105) HF: 40.05 Gy, (*n* = 61) Photon boost except 1 patient electron	BCS (*n* = 71) Mastectomy (*n* = 8) Others (*n* = 1)	CTCAE (v5.0)	Before, at the end of RT, and 6 weeks post RT
(Landoni et al. 2013) [40] Italy	Prospective	Total patients: 89 Median age: 62, range (31–79 years)	Sequoia 512 scanner (Siemens Medical Systems, USA) Probe: 8.0–15.0 MHz linear array (15L8 W)	HF: 34 Gy in 2 weeks with an electron boost of 8 Gy	BCS	CTCAEv3	11.4–85.7 months post RT (median 20.5 months)
(Garnier et al. 2017) [41] France	Prospective	Total patients: 34 Median age: 61.5, range (53–68 years) Median BMI: 23.5 Smoking (*n* = 2) Median BV: 425 mL (307–577)	1-Dermcup 2020, Atys Médical, Soucieu-en-Jarrest, France Probe: 20 MHz 2-LOGIQ7S Probe: 18 MHz	CF: 50 Gy Photon boost of 16 Gy	BCS	CTCAE v4.0	At the end of RT
(Wang et al. 2020) [33] USA	Prospective	Total patients: 109 CF = 63, median age 58, range (26–75 y), mean BMI 29.8, mean BV: 1904.5 cm^3^ (223.3–6988.29), smoking history (*n* = 14) HF = 46, median age 54.5, range (38–78 years), mean BMI 31.9, mean BV: 1961.2 cm^3^ (472.7–4113.62), smoking history (*n* = 12)	NR	CF: 50 Gy (*n* = 63) HF: 39.9 Gy (*n* = 46) Combined electron and photon boost in both groups	BCS	Breast retraction assessment	Before RT, the last day of RT, 12 weeks, and 1-year post RT
(Torres et al. 2016) [32] USA	Prospective	Total patients: 70 Mean age 57, range (26–75 years) Mean BMI 29.3 Mean BV: 1819.9 cm^3^ (223.3–6988.3) Smoking history (*n* = 14)	NR	CF: 50 Gy 10–16 Gy electron or photon boost	BCS	RTOG	Before RT, at week 6, and 6 weeks post RT
(Lin et al. 2019) [9] USA	Prospective	Total patients: 66 Mean age 56.3, range (26–75 years) Mean BMI: 29.9 Mean BV: 1919.9 cm^3^ (223.3–6988.3) Smoking history (*n* = 12)	NR	CF: 50 Gy Majority received an electron boost of 10–16 Gy	BCS	NR	Before RT, week 6 of RT, and 6 weeks, 6 months, and 1 year post RT
(Yoshida et al. 2012) [34] USA	Prospective	Total patients: 26 Mean age 55, range (42 to 74 years)	Sonix RP (Ultrasonix Medical Corporation, BC, Canada) Probe: 12-MHz linear	CF: 50.0–50.4 Gy Electron boost of 10.0–16.0 Gy	BCS	RTOG	Acute toxicity group (*n* = 8): before, during, and up to 6 months post RT Late toxicity group (*n* = 18): received one US study ≥6 months post RT
(Yoshida et al. 2011) [35] USA	Prospective	Total patients: 18 Mean age 56, range (44–74 years)	Sonix RP (Ultrasonix Medical Corporation, BC, Canada) Probe and US setting: 10 MHz linear array, 1.25 cm focal length, 4 cm depth, 72% gain, and 80 dB dynamic range	CF: 50.0 to 50.4 Gy Electron boost of 10.0 to 16.0 Gy	BCS	RTOG	6 to 92 months post RT (median, 22 months)
(Liu et al. 2010) [31] USA	Prospective	Total patients: 18 Mean age 56, range (44–74 years)	Sonix RP (Ultrasonix Medical Corporation, Richmond, BC, Canada) Probe: 12-MHz linear array	CF: 50.0 to 50.4 Gy Electron boost of 10.0 to 16.0 Gy	BCS	RTOG	6 to 94 months post RT (median, 22 months)
(Liu et al. 2008) [42] USA	Prospective	Total patients: 12	Sonix RP system Probe: 6 MHz linear array (L14–5/38) The RF data were acquired with a 20 MHz sampling frequency	CF: 50–50.4 Gy Electron boost of 10–16 Gy	BCS	NR	6 to 94 months post RT (Median, 22 months)
(Keskikuru et al. 2004) [43] Finland	Prospective	Total patients: 21	Dermascan, Denmark Probe: 20 MHz	CF: 50 Gy Boost not mentioned	BCS	Modified Dische classification for erythema at the end of RT; subcutaneous induration scoring at 1 and 2 years	Before RT, at 2.5 weeks, at the end of RT, and 1, 4, 7, 12, and 24 months post RT
(Wratten et al. 2002) [37] Australia	Prospective	Total patients: 13 age range (43–77 years)	Dermascan C, Cortex Technology, Denmark Probe: 20 MHz, an axial resolution of 60µm, a lateral resolution of 200µm, and a depth of up to 15 mm	CF: 50–64 Gy ± electron boost	BCS	Visual for the presence of oedema	Before RT and before fractions 4, 6, 9, 11, 16, 21, and 26
(Wratten et al. 2000) [38] Australia	Prospective	Total patients: 11 age range (35–72 years)	Dermascan C, Cortex Technology, Denmark Probe: 20 MHz, an axial resolution of 60µm, a lateral resolution of 200µm, and a depth of up to 15 mm	CF: 44–50 Gy Electron boost of 10–20 Gy	BCS	Visual for the presence of oedema	Different time assessments for different patients: baseline, during, and post RT, including one patient up to 22 years post RT
(Warszawski et al. 1998) [44] Germany	Prospective	Total patients: 29 Median age: 58, range (39–72 years)	Taberna pro Medicum (DUB 20) (Luneburg, Germany) Probe: Digital linear 20 MHz, axial, and lateral resolutions of 70 and 150 mm, respectively	CF: 46–50 Gy 10 Gy electron boost for patients with BCS	BCS (*n* = 23) + Mastectomy (*n* = 6)	RTOG	≤3 months in 18 patients and 6–135 months post RT in 11 patients
(Schack et al. 2016) [45] Denmark	Prospective	Total patients: 15 Median age 66, range (44–75 years) Median BV: 715 mL (177–1627)	DermaScan C from Cortex Technology ApS, Denmark Probe: 20 MHz with a maximum depth of 6 mm	40 or 50 Gy (no details)	BCS	LENT-SOMA Scale	Baseline, post RT (median 3.0 years (1.0–4.6 years))
(Wong et al. 2011) [46] Singapore	Prospective	Total patients: 32 Median age: 52.5, range (37–68 years)	Sequoia^®^ 512 scanner (Siemens Medical Systems, USA) Probe: linear array (15L8 W), 14 MHz centre frequency, and a maximum depth of 80 mm	46–50 Gy No boost	Mastectomy	RTOG	16–39 months post RT (median, 27.5 months)
(Wratten et al. 2007) [36] Australia	Prospective	Total patients: 54 Median age: 55, range (31–74 years)	Dermascan C, Cortex Technology, Smedevaenget, Denmark Probe: 20 MHz	CF: 50–66 Gy Electron boost 10–20 Gy (*n* = 21)	BCS	NR	Baseline, weekly during RT, at 2 weeks, 6 weeks, 4 months, 6 months, 12 months, and 24 months post RT

Abbreviations: RT = radiotherapy, US = ultrasound, NR = not reported, BV = breast volume, RF = radiofrequency, RTOG = Radiation Therapy Oncology Group, CTCAE = Common Terminology Criteria for Adverse Events, CF = conventional fractionation, HF = hypofractionation, LENT-SOMA = late effects normal tissues-subjective, objective, management, analytic.

**Table 3 ijerph-19-13439-t003:** Ultrasound skin toxicity findings of the irradiated breast compared with the nonirradiated or pre-RT breast.

Author(s), Year	Measured Parameters and Locations	Ultrasound Skin Findings	Main Findings
(Borm et al. 2021) [39]	Skin thickness Locations: at 12:00, 3:00, 6:00, and 9:00 around the mamilla	Healthy breast: 1.7–1.8 mm at all timepoints for both groups Treated breast: HF: 2.3 mm before RT, 2.4 mm at the end of RT, and 2.5 mm post RT CF: 2.3 mm before RT, 2.3 mm at the end of RT, and 2.5 mm post RT	- ↑ skin thickness compared to the HB before and after RT - Skin thickness ↑ after RT in both groups - No significant difference between HF-WBI and CF-WBI both at the end and following RT
(Landoni et al. 2013) [40]	Skin thickness Locations: the irradiated breast, boost region, and corresponding positions on the untreated breast	Mean skin thickness: Irradiated breast: 2.13 ± 0.72 mm Contralateral site: 1.61 ± 0.29 mm Boost region: 2.25 ± 0.79 mm. Contralateral site: 1.63 ± 0.33 mm	- Significant difference in both examined regions - No significant difference between the boosted area and the irradiated breast
(Garnier et al. 2017) [41]	Dermal thickness Locations: the irradiated breast, boost region, and corresponding positions on the untreated breast	Median dermal thickness (mm): Irradiated skin: 1.7 [1.4–2.1] Contralateral site: 1.3 [1.0–1.5] Boost region: 1.7 [1.4–2.1] Contralateral site: 1.2 [1.0–1.4]	- ↑ dermal thickness compared to the HB - ↑ breast volume, young age, and dark skin phototype were associated with more severe skin toxicity
(Wang et al. 2020) [33]	Mean epidermal thickness STRA Locations: the four quadrants of both breasts	Mean STRA: Baseline CF: 1.3, HF: 1.3 During RT CF: 1.5, HF: 1.4 12 weeks post RT CF: 1.6, HF: 1.5 1 year post RT CF: 1.5, HF: 1.5	- STRA was ↑ before RT - Mean STRA was ↑ post RT compared to baseline - Breast volume and supraclavicular irradiation were associated with the most significant changes in breast asymmetry
(Torres et al. 2016) [32]	Mean epidermal thickness and STRA Locations: the four quadrants and boost region in both breasts	STRA: Baseline: 1.27 (SD 0.29). During RT: 1.52 (SD 0.46) 6 weeks post RT: 1.6 (SD 0.46).	- Mean ↑: Before RT: 27% During RT: 25% Post RT: 33% - Mean STRA 6 weeks after RT was significantly larger than baseline - ↑ breast volume is a consistent patient-related factor associated with ↑ epidermal thickening secondary to RT
(Lin et al. 2019) [9]	Epidermal thickness STRA Locations: the four quadrants of both breasts	Mean STRA: Baseline: 1.28 ± 0.31. At week 6: 1.55 6 weeks post RT: 1.62 6 months post RT: 1.65 ± 0.41 1 year post RT: 1.44 ± 0.38	- ↑ STRA post RT compared to baseline - Significant increase at 6 months (absolute mean ↑ of 65%, SD 0.054) - 1 year post RT (absolute mean ↑ of 44%, SD 0.048) - In MVA, ALND, longer interval between surgery and RT, ↑ baseline STRA, and Caucasian race predicted more severe changes in STRA at one year compared to baseline (all *p* < 0.05)
(Yoshida et al. 2012) [34]	Ultrasound RF data: skin thickness PCC of the hypodermis Locations: 12:00, 3:00, 6:00, 9:00, and tumour bed of both breasts	Dermal toxicity: 28.5% ± 26.6% for RTOG = 0 and 69.7% ± 39.7% for RTOG = 1 or 2 Hypodermal toxicity: 5.4% ± 35.8% for RTOG = 0 and 19.2% ± 26.2% for RTOG = 1 or 2	- Significant dermal toxicity changes in the acute and late toxicity groups - Significant hypodermal toxicity changes in the early but not in the late group
(Yoshida et al. 2011) [35]	Ultrasound RF data: skin thickness and PCC Locations: 12:00, 3:00, 6:00, and 9:00 positions of both breasts	The average skin thickness: Treated breast: 2.61 mm (1.53–3.65 mm) Untreated breast: 2.05 mm (1.66–2.41 mm) Average PCC: Treated breast: 0.28 (range: 0.21–0.41) Untreated breast: 0.41 (range: 0.03–0.52)	- Significant differences - 27.3% mean ↑ in skin thickness, 34.1% mean ↓ in PCC
(Liu et al. 2010) [31]	Ultrasound RF signals: skin thickness and PCC Locations: 12:00, 3:00, 6:00, and 9:00 of both breasts	Skin thickness range: Untreated breasts: 1.66–2.41 mm Treated breasts: 1.53–3.65 mm PCC range: Untreated hypodermis: 0.03 to 0.52 Treated hypodermis: 0.21 to 0.41	- Significant changes in both parameters - Average skin thickness ↑ by 27.3%, and the PCC ↓ by 31.7%
(Liu et al. 2008) [42]	Ultrasound RF signals: skin thickness and PCC Locations: irradiated breast and nonirradiated breast	Average skin thickness: Irradiated skin: 3.3 ± 1.4 mm (2.01 to 5.82 mm) Nonirradiated skin: 2.2 ± 0.4 mm (1.93 to 2.75 mm) Average PCC: Irradiated hypodermis: 0.18 ± 0.08 (0.01 to 0.36) Nonirradiated hypodermis: 0.27 ± 0.10 (0.10 to 0.42)	- Significant changes in both parameters - The average skin thickness ↑ by 40%, and the average PCC↓ by 35%
(Keskikuru et al. 2004) [43]	Skin thickness of induced suction blisters Location: in the upper medial quadrant of both breasts	The mean skin thickness of the Irradiated breast: Before RT: 1.9 mm, 4 months: 2.1 mm 7 months: 2.00 mm 1 year: 1.9 mm 2 years: 1.7 mm Nonirradiated breast: <1.8 mm at all timepoints	- Significant changes from 2.5 weeks of RT, peaked at around 4 months, then declined until 2 years - 9% ↑ in skin thickness in the operated breast before RT
(Wratten et al. 2002) [37]	Skin thickness Locations: 4 cm medial and lateral to the nipple in both breasts	Mean skin thickness of: Medial treated breast 2.23 mm Lateral treated breast 1.91 Medial untreated breast 1.38 mm Lateral untreated breast 1.16 mm	- The treated breast skin is overall thicker than the untreated breast - Thickening is evident before RT - The medial aspect is thicker than the outer aspect in both breasts
(Wratten et al. 2000) [38]	Total cutaneous thickness Locations: 4 cm medial and lateral to the nipple in both breasts	The mean total cutaneous thickness: Treated breast: 2.71 mm (range 1.42–4.66 mm, SD 0.83 mm) Untreated breast: 1.35 mm (range 0.84–1.82 mm, SD 0.21 mm)	- Significant changes - The medial aspect of the breast was thicker than the lateral aspect in both breasts - ↑cutaneous thickness before RT in those patients with axillary dissection
(Warszawski et al. 1998) [44]	Entry echo of the skin Corium (dermal) thickness The echogenicity of the upper and lower corium Structure of the border between the corium and subcutis Locations: at the edge between the upper quadrants 2–3 cm above the mammilla in both breasts	Mean corium thickness (µm): Nonirradiated skin: 1683 ± 308 Early reactions: 2683 ± 721 Late reactions: 2307 ± 934 The mean echogenicity of the upper and lower corium: Nonirradiated skin: 3.63 ± 1.58 to 5.04 ± 1.56 Early reactions: 1.90 ± 1.37 to 1.93 ± 0.76 Late reactions: 2.32 ± 0.88 to 3.33 ± 1.41 Unsharp dermis-subcutaneous border: Nonirradiated skin: 5/31, 16.1% Early reactions: 15/44, 34.1% Late reactions: 8/21, 38.1%	- Significant changes in the early and late reactions for the thickness and echogenicity of the dermis - Nonsignificant changes for the entry echo and poor visibility of the dermis-subcutaneous border - No significant difference between early and late reactions except for the echogenicity of the lower corium
(Schack et al. 2016) [45]	Dermis thickness Locations: 3 cm from the areola in all four quadrants of both breasts	The mean dermis thickness: Untreated breast: 1.26 mm (95% CI 1.08–1.44) Irradiated breast: 2.22 mm (95% CI 1.78–2.66) The mean difference: 0.96 mm (95% CI 0.50–1.42)	- Statistically significant changes
(Wong et al. 2011) [46]	Skin thickness Locations: 9 points within the medial, central, and lateral areas of both breasts	Mean skin thickness (mm): Irradiated Rt chest wall: 0.1712 Nonirradiated Rt side: 0.1845 Irradiated Lt chest wall: 0.1764 Nonirradiated Lt side: 0.1835	- A significant difference in ↓ skin thickness of the irradiated chest wall compared to the non-irradiated chest - The findings indicated chronic skin reactions - The medial aspect was consistently thicker than the lateral aspect
(Wratten et al. 2007) [36]	Mean epidermal thickness Locations: 4 cm medial and lateral to the nipple in both breasts	Treated breast: Baseline: 1.9–2.3 mm During RT: 1.9–2.5 mm 4–6 months: 2.3–3 mm 1–2 years: 1.5–2.5 mm Untreated breast: 1.3–1.5 mm at all timepoints	- Peaked at 4–6 months post RT and mostly returned to baseline levels by 12 months post-RT - ↑ epidermal thickness before RT in patients with ALND - The thickness of the epidermis was greater medially in both breasts - Irradiation of the breast causes little cutaneous oedema in the absence of axillary dissection or nodal irradiation

Abbreviations: RT = radiotherapy, HB = healthy breast, RF = radiofrequency, CF-WBI = conventional fractionated whole-breast irradiation, HF-WBI = hypofractionated whole-breast irradiation, ALND = axillary lymph node dissection, PCC = Pearson correlation coefficient, MVA = multivariate analysis, STRA = skin thickness ratio.

**Table 4 ijerph-19-13439-t004:** The US skin changes and the association with clinical assessments.

Author(s), Year	US Changes in Parameter		Association with Clinical Assessment	Notes
	Early (≤3 Months)	Late (˃3 Months)		
(Borm et al. 2021) [39]	↑ skin thickness compared to the HB		At the end of RT: no significant difference in skin thickness but significant difference (*p* = 0.03) in the CTCAE score 6 weeks post RT: no significant difference in skin thickness or CTCAE score (*p* = 0.39)	HF is associated with a lower degree of acute RD compared to CF at the end of treatment CTCAE scores and US measurements do not reliably reflect the patient’s perception
(Garnier et al. 2017) [41]	↑ dermal thickness compared to the HB		The mean relative ↑ in dermal thickness in irradiated skin (RIDTIS) was greater for grades 2 and 3 than 1: 0.53 vs. 0.29 mm (*p* = 0.023)	US of dermal thickness may be a reliable tool to quantify acute RD
(Wang et al. 2020) [33]	Increased STRA compared to baseline	Increased STRA compared to baseline	A significant association between STRA and breast asymmetry (*p* = 0.02, 0.04, <0.01 at baseline, 12 weeks, and 1 year post RT, respectively)	HF is associated with better long-term cosmetic outcomes Supraclavicular nodal irradiation and CF are associated with worse cosmetic outcomes 1 year post RT
(Torres et al. 2016) [32]	Significant difference compared to baseline (*p* < 0.001) and the end of RT (*p* = 0.03)		Correlated with RTOG Mean STRA is ↑ in patients with grade 2 than grade 0 at the end of RT (*p* = 0.001) and 6 weeks post RT (*p* < 0.03)	RT had a synergistic effect with lymph node surgery on breast skin thickening
(Lin et al. 2019) [9]	Significant changes compared to baseline (*p* < 0.001)	Significant changes compared to baseline (*p* < 0.001)	NR	ALND has a long-term impact on breast skin thickening
(Yoshida et al. 2011) [35]		Significant difference of skin thickness (*p* < 0.001) and PCC (*p* < 0.001).	PCC correlated with RTOG late toxicity, but skin thickness did not (↑38.4% for RTOG grade 0, 23.8% for grade 1, and 31.1% for grade 2 toxicity); *p*-value NR	Quantitative US is an objective tool that assesses RT-induced tissue injury, which may improve patients’ quality of life
(Yoshida et al. 2012) [34]	Significant dermal (*p* < 0.0001) and hypodermal toxicity (*p* = 0.0027)	-Significant dermal toxicity (*p* < 0.05) -Not significant hypodermal toxicity (*p* = 0.22)	Late toxicity assessments correlated with RTOG (Patients with RTOG grade 1 or 2 have greater US toxicity changes than patients with RTOG grade 0, *p* = 0.04 for dermal toxicity and *p* = 0.22 for hypodermis toxicity)	Early and late radiation-induced effects on normal tissue can be reliably assessed using the quantitative US
(Keskikuru et al. 2004) [43]	Significant changes in skin thickness (*p* < 0.05, *p* < 0.01 at different timepoints)	Significant changes (*p* < 0.05, *p* < 0.01 at different timepoints) until one year then declined	No significant correlations between the skin thickness and the score of erythema or subcutaneous induration	Increased collagen synthesis is associated with oedema resulting from radiation-induced damage to skin microvasculature
(Wratten et al. 2000) [38]	Significant changes (*p* < 0.001)	There was a persistent ↑ in cutaneous thickness in the treated breast	The most prominent visual breast oedema exhibited the greatest total cutaneous thickness (*p*-value NR)	HFUS can quantify cutaneous breast oedema accurately
(Wratten et al. 2002) [37]	No obvious skin thickness changes during RT (*p*-value NR).		The most marked cutaneous thickness was in patients with obvious visible breast oedema before RT (*p*-value NR)	HFUS is not an ideal, sensitive, and quantitative measure of acute RD in this group of patients
(Wratten et al. 2007) [36]	A minor ↑ in epidermal thickness (*p*-value NR)	Significant changes (*p* = 0.000 with or without level 2 nodal dissection)	NR	The utility of HFUS is in a research setting when assessing interventions that aim to reduce breast oedema
(Warszawski et al. 1998) [44]	Significant changes in the dermis thickness and echogenicity (*p* < 0.001) Nonsignificant changes in the structure of the dermis-subcutis border (*p* = 0.07)	Significant changes in the dermis thickness (*p* = 0.0018) and echogenicity (*p* < 0.001 for lower dermis, *p* = 0.0027 for upper dermis) Nonsignificant changes in the structure of the dermis-subcutis border (*p* = 0.08)	There were discrepancies between the clinical and US assessments, mainly in the late reactions (K = −0.13, Pearson’s correlation) Early skin reactions: structural changes could be recorded by US evaluation much earlier than visible reactions by the naked eye	High resolution 20 MHz US is noninvasive, quantitative, and reproducible for assessing early and late skin reactions US skin changes depend on the time interval between completion of RT and US evaluations
(Landoni et al. 2013) [40]		Statistically significant difference (*p* < 0.001)	US assessments were in agreement with clinical assessments A significant direct correlation was found between the increment in skin thickness with fibrosis (grade ≥ 1) in the irradiated breast (*p**-*value = 0.0236) and the boost region (*p**-*value = 0.0164)	Late cutaneous reactions can be reliably assessed by US
(Liu et al. 2008) [42]		Significant skin thickness (*p* = 0.005) and Pearson coefficient (*p* = 0.02) changes	NR	US technique is noninvasive and feasible to detect and quantify radiation-induced skin changes
(Liu et al. 2010) [31]		Significant skin thickness and Pearson coefficient changes (*p* < 0.001)	US evaluations were consistent with RTOG scores Skin thickness correlated with RTOG late subcutaneous toxicity, and PCC correlated with late skin toxicity (*p*-value NR)	The quantitative US is noninvasive and objective for assessing radiation-induced changes to the skin
(Schack et al. 2016) [45]		Significant differences (*p* = 0.0003)	The highest mean difference in dermis thickness (1.61 mm (95% CI 0.41–2.82) was in patients with clinical oedema and grade 2 induration (*p* = 0.02)	HFUS evaluation of the skin is not part of large-scale follow-up routines in assessing radiation-induced morbidity
(Wong et al. 2011) [46]		Significant skin thickness changes of the Rt chest (*p* = 0.007) and Lt chest (*p* = 0.025)	US measurements correlated with RTOG Patients with grade 2 acute skin toxicity presented with thinner skin (mean skin thickness 0.1720 mm) compared to patients with grade 1 (0.1879 mm) (*p* = 0.006)	HFUS can be utilised to document quantitative skin changes following postmastectomy RT

Abbreviations: NR = not reported, RT = radiotherapy, US = ultrasound, HB = healthy breast, HFUS = high-frequency ultrasound, RD = radiation dermatitis, RTOG = Radiation Therapy Oncology Group, CTCAE = Common Terminology Criteria for Adverse Events, CF = conventional fractionation, HF = hypofractionation, ALND = axillary lymph node dissection, PCC = Pearson correlation coefficient, STRA = skin thickness ratio.

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
