# Peer review of "Ultrasonographic Evaluation of Skin Toxicity Following Radiotherapy of Breast Cancer: A Systematic Review"

_ijerph, 2022, doi:10.3390/ijerph192013439_

Round 1

Reviewer 1 Report

The authors carried out an interesting review on the ultrasonographic evaluation of skin toxicity after RT of breast cancer. Overall, the paper is clear and well written. I have made few suggestions that the authors may want to consider in revising their paper.

1.Abstract

1.1. Please check the guidelines of IJERPH. It should be without headings and within a 200-word limit.

2.Introduction

2.1. The introduction and the review question are clear and well written.

3. Methods

3.1. It is not clear if both title+abstract and full text selection were done by F.A.H. and reviewd by H.A.M. and N.Y.

4.Results

4.1. Please consider adding the reference of the only study with good quality in the text and adding more information on quality and possible biases of the included studies.

4.2. Please consider using the updated PRISMA flow diagram for Figure 1.

4.3. Check the title of Table 4.

4.4. Table 4: should the empty cells be NR?

4.5. “Despite the subjectivity of the clinical assessments and scoring scales, they are still the commonest toxicity evaluation during and following RT. Comparing with clinical assessments should be considered for any objective/quantitative technique (Yoshida et al. 342 2012).” Perhaps this consideration belongs more to the discussion than to the results

5.Discussion

5.1. I think that the review question and the aims in the first line of discussion are similar but do not have the same meaning. Please check and consider rephrasing to clarify your aim

Author Response

Dear reviewer, 

Thank you very much for the comments and suggestions. Both comments and suggestions tremendously improve the quality of the manuscript. 

Please find response in the attached file. 

Thank you. 

Reviewer 2 Report

This study is well designed, and comprehensively written with good efforts of authors. Authors should be commended for such efforts. If authors are radiologists, I suggest authors can show some figures of their own relating to the results. Because many oncologists who are not radiologists can feel difficulty to understand dermatologic condition in numbers and texts. 

Author Response

(The authors gave the same response as above.)
